# “Food Is Our Love Language”: Using Talanoa to Conceptualize Food Security for the Māori and Pasifika Diaspora in South-East Queensland, Australia

**DOI:** 10.3390/nu14102020

**Published:** 2022-05-11

**Authors:** Heena Akbar, Charles J. T. Radclyffe, Daphne Santos, Maureen Mopio-Jane, Danielle Gallegos

**Affiliations:** 1Woolworths Centre for Childhood Nutrition Research, Queensland University of Technology, South Brisbane 4101, Australia; h.akbar@qut.edu.au (H.A.); c_radclyffe93@hotmail.com (C.J.T.R.); 2School of Exercise and Nutrition Sciences, Queensland University of Technology, Kelvin Grove 4059, Australia; 3Pasifika Young Peoples Well-Being Network (PYPWN), School of Public Health and Social Work, Queensland University of Technology, Kelvin Grove 4059, Australia; 4Good Start Program, Child and Youth Community Health Services, Children’s Health Queensland, South Brisbane 4101, Australia; daphne.santos@health.qld.gov.au; 5Elder of the Papua New Guinea Community, Queensland 4000, Australia; mmopiojane@me.com

**Keywords:** food security, culture, food sovereignty, monitoring, Pacific Islands, Māori, Melanesian, Micronesian, Polynesian

## Abstract

Queensland is home to the largest diaspora of Māori and Pasifika peoples in Australia. They form an understudied population concerning experiences and challenges of food insecurity. This community co-designed research aims to explore the conceptualization of household food security by Māori and Pasifika peoples living in south-east Queensland. Participatory action research and talanoa were used to collect and analyse forty interviews with leaders representing 22 Māori and Pasifika cultural identities in south-east Queensland. Eight key themes emerged that conceptualise food security as an integral part of the culture and holistic health. These themes included: spirituality, identity, hospitality and reciprocity, stigma and shame, expectations and obligations, physical and mental health and barriers and solutions. Addressing food insecurity for collectivist cultures such as Māori and Pasifika peoples requires embracing food sovereignty approaches for improved food security through the co-design of practical solutions that impact social determinants and strengthen existing networks to produce and distribute affordable and nutritious food.

## 1. Introduction

Food security exists “when all people at all times have physical, social, and economic access to sufficient, safe and nutritious food to meet their dietary needs and food preferences for an active and healthy life” [1]. Being able to feed yourself and your family with food that is of sufficient quantity and quality is a fundamental human right [2]. In high-income countries, such as Australia, food is often available in sufficient quantities, but access to a diet of sufficient quality to sustain health can be compromised [3]. A recent review has also highlighted that culture, including but not limited to religiosity, gender, perceptions of stigma and food sharing norms, potentially impacts on every element of food security [4].

Food insecurity at the household level has serious implications for the physical and psycho-social health and well-being of adults and children. Food insecure adults and children have increased healthcare use and presentations to emergency departments [5,6,7,8]. Adults experiencing household food insecurity (HFI) have a greater risk of chronic conditions (diabetes, heart disease) and mortality [9,10,11,12] as well as poor mental health [13,14,15]. In children, HFI has been associated with compromised mental health [16], as well as developmental, cognitive and behavioural (internalising and externalising) anomalies [17]. In parts of Australia, from one in four to one in two households experience food insecurity on a regular basis [7,18,19,20,21,22]. The primary determinant of HFI is low income combined with the high cost of living, including unaffordable housing, utilities, healthcare, education, and food [3,23,24,25]. Other contributing factors include the number of adults and children in a household, mental and physical health of adults and children, psychological stress, adverse childhood experiences, domestic and family violence and experiences of racism [26,27,28,29,30]. 

Māori and Pasifika peoples currently make up an estimated 102,320 of the Queensland population, which is the largest Māori and Pacific Islander population in Australia [31]. Due to a complex interplay between cultural and social determinants as well as migration conditions, Māori and Pasifika peoples living in Queensland are at high risk of food insecurity. The next section provides background into the unique position of the Māori and Pasifika diaspora living in Australia.

### 1.1. Māori and Pasifika in Australia

The term ‘Pasifika’ is used in this study to describe the intergenerational diaspora of migrants who belong to and identify with Oceania which encompasses all Pacific Island nations that constitute Micronesia, Melanesia and Polynesia [32]. The term acknowledges and celebrates the layered and rich diversity of this oceanic continent; however, it recognises that there are sufficient commonalities across beliefs, values, and practices [33]. ‘Pasifika’ has been used in other Australian health studies and accepted by some communities to represent the Māori (tangata whenua—“people of the land”) of Aotearoa (New Zealand) [32,33]. However, the communities in this research requested the use of ‘Māori and Pasifika’ separately in recognition that many Pasifika have New Zealand citizenship, but Māori are the First Nations people of Aotearoa.

Māori and Pasifika communities in Queensland are well connected and work collectively through a rich network of cultural, sporting, religious, educational, and support organisations [33,34]. The extended network is the location for the cultivation and enactment of identity, spirituality, social principles, and responsibilities [35,36]. Despite these considerable strengths, Māori and Pasifika peoples are impacted by a combination of pre-existing social and cultural determinants and the precarity engendered by the immigration stance of the Australian government. This means that Māori and Pasifika peoples are more likely to work in unskilled or semi-skilled positions and be at risk of irregular, casualised employment due to racism, language ability and low education attainment [37,38]. There is a lack of recent data, but in Australia and Queensland specifically, Māori and Pasifika peoples are more likely to be hospitalised and die prematurely from chronic conditions such as diabetes [39,40]. The high risk and prevalence of chronic conditions are exacerbated by poor diet quality [41,42]. Diet quality can be difficult to maintain due to cultural food insecurity, where the food preferences of the community cannot be realised due to those foods being unavailable or too expensive [43].

One of the most significant developments in Māori and Pasifika migration between New Zealand, Australia and Pacific Island nations was the establishment of the Trans-Tasman Travel Arrangement (TTTA) in 1973. The TTTA is firmly rooted in Australian attempts to reduce perceived “back door” migration to Australia [36,44]. Changes to the TTTA have resulted in visa holders after 2001 being ineligible for social security with only limited access to the National Disability Scheme, public and emergency housing, transport concessions and other assistance [45]. The system creates a lack of equity, high rates of unemployment, under-employment and precarious employment contributing to low incomes, all of which contribute to food insecurity. 

Food insecurity is therefore potentially a significant issue for this population due to the financial hardships generated by the inability to access social protection, public housing and other social services as well as increased vulnerability to the escalating cost of living and shocks such as adverse weather events. This is further exacerbated by the sending of remittances to Island communities [46]. All of these factors contribute to family breakdown, over-representation in the justice system [36], and overall poorer health [40]. It is therefore essential to explore how food insecurity is conceptualized in this community in order to co-design strategies in the future. 

### 1.2. Aims

The aim of this research was to explore how Māori and Pasifika communities living in south-east Queensland, conceptualise and perceive food security. The results will enable an in-depth understanding of the significance of food beyond physical health and contribute to co-designing solutions to ensure food security for Māori and Pasifika communities.

## 2. Materials and Methods

This research is reported using the Standards for Reporting Qualitative Research (SRQR) guidelines (see Appendix A).

### 2.1. Design

This qualitative research was underpinned by the principles of a rights-based approach to food, focusing on the alteration of conditions and environments to enable people to take an active role in procuring food [47]. Participatory action research (PAR) utilising co-design and talanoa methods was employed as the theoretical and cultural framework in the collection and analysis of data. ‘Talanoa’ is a Pasifika method/methodology describing a process of enabling critical discussions and knowledge co-construction. ‘Tala’ means to inform or relate and ‘noa’ means to share as an exchange without a rigid framework [48].

The project was led by a Steering Committee (SC) including Māori and Pasifika members living in south-east Queensland, a Pasifika university researcher (HA) and an experienced university researcher (DG). The SC was guided by a co-developed terms of reference that outlined expectations and parameters of engagement. The SC decided the data to be collected and analysed by community researchers (CR) drawn from the Pasifika communities. Six Elders and two young researchers (<30 years) volunteered from the SC and an additional Elder researcher and five young researchers were recruited from the communities and the Pacific Young Peoples Wellbeing Network (PYPWN). Each young researcher was paired with an Elder researcher to optimise the knowledge exchange and to promote intergenerational conversations regarding culture and food. All CRs received training in culturally safe data collection, ethics (including confidentiality), data management and analysis within the context of talanoa and were remunerated as research assistants. 

### 2.2. Participants and Recruitment

The SC brainstormed members of the Māori and Pasifika communities who held positions of high esteem or were known leaders, had advocated on food insecurity or were developing interventions related to food security (e.g., food pantries, gardens). Key recruitment considerations included capturing a diverse representation from all Pacific Island nations (irrespective of the size of the community), from multiple generations and those born outside and in Australia. Community members were approached initially by an SC member, either by telephone or email, to gauge interest to participate in the study. Once they had agreed, participant information was sent and follow up was made by the HA. This allowed community members to agree or decline to participate and this process reduced the risk of perceived coercion. HA emailed the participant information, consent, and the interview process with each participant and answered any questions regarding the research. Finally, participants were able to declare a preference for an interviewer allowing the agency to request or decline a particular CR. This enabled the participant to feel comfortable if sensitive material was to be shared. Interview day, time, location and mode were organised by the CRs and their assigned participant. 

### 2.3. Data Collection

The SC iteratively developed the talanoa questions which were further refined in consultation with the CR. The questions and talanoa were then piloted with members of the SC to ensure they were clearly understood, culturally appropriate and elicited the information expected based on the research questions. The interview guide is located in Appendix A. 

Prior to each talanoa, the CR went through the participant information again, answering any additional questions and gained written or verbal consent. Cultural sensitivity and respect through talanoa formed the underpinning guiding value in this research process [48]. Each participant received an AUD 30 dollar grocery voucher as koha or a gift for participation.

Data collection occurred over a three-month period from September to November 2020 and interviews were conducted face-to-face or via the online video communication platform Zoom (©2021 Zoom Video Communications, Inc., San Jose, CA, USA), based on participant preference. Talanoa sessions took between 90 to 120 min. All interviews were audio-recorded and transcribed using Otter ai 2.0 Software (Otter.ai. Software Development, Mountain View, CA, USA). Each CR team uploaded the recording onto a secured shared drive, accessible only to the university researchers. Participants were given the opportunity to read a copy of their transcript and make changes. Each interview transcript was reviewed by the lead researchers for verbatim accuracy against the recording, making sure that any words in the language were captured correctly. Words that were incorrectly transcribed were reviewed with the CR and SC and edited accordingly.

### 2.4. Data Co-Analysis

Figure 1 summarises the data analysis process.

Due to the volume and depth of data collected and the logistics of involving 16 researchers in the data analysis, the SC and university researchers (HA and DG) co-developed a contact summary sheet (CSS) [49]. A CSS is a single sheet containing a series of focusing or summarizing questions about a particular interview (see Appendix A). The CSS was iteratively developed with the SC and CR and included verbatim quotes. Community and university researchers independently listened to and read one interview audio recording and transcript to complete the CSS. Any differences were discussed to develop a shared understanding of the material. University researchers completed the remaining transcripts. The CSS was used in conjunction with the interview transcripts for coding and thematic analysis. To aid in interpreting the large datasets, each transcript was uploaded to an online wordle tool (www.edwordle.net accessed on 29 April 2021). The wordle was set to 250 words, to group similar words and exclude stop words (Figure 2).

#### Thematic Analysis

Line-by-line coding and inductive thematic analysis were undertaken by the university researchers using the steps outlined by Braun and Clarke to develop a preliminary codebook [50]. The preliminary codebook, transcripts, CSS and wordles were used in a talanoa workshop with CR. This workshop aimed to further develop the analysis framework and identify preliminary themes. Community researchers (*n* = 6) were divided into three groups, with the university researchers (*n* = 2) joining two of the groups. Transcripts were divided between these groups according to age; that is, one group received CSS and transcripts for the older generation (O = older), one for the middle generation (M = middle) and one for the younger generation (Y = younger). This was to highlight any generational differences in data. Each group received two interviews that they conducted, and four interviews conducted by other teams. Community researchers then used butcher’s paper and post-it notes to cluster ideas into themes. The final step involved the CRs presenting their analysis to the whole group highlighting the core themes which were discussed and then formed the preliminary codebook. 

The second stage of analysis involved an additional talanoa workshop with Elder CR and the university researchers. The Elder CR used the preliminary codebook and the remaining CSS with exemplar quotes to further extend and verify the themes. These themes were used to develop a free-form mind map that further facilitated analysis. This technique was used to facilitate the connection of concepts and to generate associations between ideas [51] and involved centralising the issue (‘food’) and then branching off related concepts and headings [52]. These branches were further subdivided and/or related back to other branches as analysis progressed, resulting in the development of the mind map. After this workshop, the two university researchers met to continue the mind map using colour coding to denote generational differences and asterisks to connect back to the themes identified in the first workshop.

The mind map and generated themes were presented (including in report form) to the SC and CR. The SC provided feedback on the wording, framing and ordering of the themes. Using multiple researchers and an analysis approach that involved multiple methods, including those that are highly visual, allowed for triangulation of data. A constant comparative approach that was managed by the university researchers ensured that data integrity was maintained.

### 2.5. Reflexivity 

As is the practice in qualitative research and to acknowledge the shared expertise a reflexivity statement [53] is provided here. The authorship team comprises four members who identify as Pasifika (Akbar, Radclyffe, Santos, Mopio-Jane) of mixed cultural backgrounds. Four authors have been university trained in public health, nutrition, and anthropology. MM is an Elder of the Papua New Guinea community and a broadcaster for ethnic radio. Gallegos identifies as Australian and is a white, Anglo-Celtic, middle-class cis-female who has worked in academia for 15 years. A full listing of the SC and CR is provided in Appendix A. At multiple points in this research project, community researchers were asked to reflect on their own positionality and on the process. This process was multi-layered with Elder and young researchers debriefing with each other, as well as CR reflecting with university researchers during all stages of the research. 

### 2.6. Ethical Considerations

This research had ethical clearance from the Queensland University of Technology Human Research Ethics (#2000000381). The SC co-designed and reviewed all participant information and consent documents and all research processes to ensure their cultural integrity. All participants provided informed written or verbal consent. Any potential ethical issues arising during the interview were discussed allowing the participants to renegotiate informed consent based on the changing nature of the inquiry. Any cultural, spiritual and personal concerns about disclosure and anything in relation to cultural and religious beliefs were also discussed and clarified with participants before obtaining consent to disclose or use any sensitive information. For example, many Elders are secret keepers of specific cultural foods which have deep spiritual meaning. It was made clear that they were under no obligation to disclose this information.

## 3. Results

Fifty-nine community members were invited to participate, 15% (*n* = 9) declined and 17% (*n* = 10) were uncontactable or did not respond. Consequently, forty talanoa were undertaken with a total of 49 community members representing multiple Māori and Pasifika cultural identities (Table 1). Interviews were conducted with 34 individuals, 10 couples and there was one (*n* = 5) multi-Pasifika group talanoa. Approximately half (45% *n* = 18) were members of the older generation (>50 years), and the remaining participants were equally divided between the middle generation (*n* = 11, 30–50 years) and the younger generation (*n* = 11, <30 years). Two-thirds (67%) of participants identified as women.

From the talanoa, eight key themes emerged. In keeping with the Māori and Pasifika story-telling tradition, the themes are represented metaphorically and visually in Figure 3. Taro was chosen as it was described as a common staple food among most of those interviewed. Each theme is further discussed below with selected quotes. Quotes are designated by community group/cultural identification and generation (O; M; Y). Solutions and barriers to food insecurity will not be discussed in detail in this paper.

It is important to note that these themes are not mutually exclusive, and they overlap and provide a rich description of food, identity and holistic health in Māori and Pasifika cultural groups living in south-east Queensland. The themes describe Māori and Pasifika as a collective but this in no way implies homogeneity across the island nations. Food as a cultural object is constantly evolving and its meaning varies not only from nation to nation but also from family to family and from individual to individual. We have where possible explored in depth the similarities and have attempted to highlight the differences. However, it has not been possible to give due acknowledgement to the deep and nuanced differences across cultural groups and generations. 

### 3.1. Identity—Food Is Central to Cultural Identity and Connects People, Families and Communities


*In Chuukese there is a saying ‘food is bone’ which explains the centrality of food to life, to cultural, familial and communal shared experiences. In the way that bone provides the skeletal structure of the human body, so food is the framework to a holistic life and shared social and cultural experiences” (O_Federated States of Micronesia).*


Food plays an important social and cultural role in all Pacific communities, often beyond nourishment. Forming a central component of Pasifika cultures and everyday living, food plays a vital role in keeping traditional practices alive. In traditional contexts, food is a means of maintaining societal norms and practices and affirms one’s identity and sense of belonging. Food and its sharing is the social cement that provides a framework for shared social, cultural and communal experience and includes nurturing and sustaining individual, family and communal relationships, particularly intergenerationally for Māori and Pasifika peoples where food has and continues to be core to Pasifika concepts of love, reciprocity and respect. 


*It’s sort of like a love language in ways… giving and receiving food. For example, when my father was in hospital and my mom was in town at the time, it was the fact that my friends showed up with food for us. That was a big deal for her (Y_Papua New Guinea).*


Every Pasifika culture has specific protocols on how certain types of foods are prepared, served and to who and by whom as a way of maintaining customs. It is through food that familial and community connections are forged, maintained and strengthened. Traditional foods are pivotal to this identity formation and are valued not only for their nutrition but also for learning about one’s culture. Customary practices regarding how food is prepared, stored, distributed and consumed, is passed down through the generations. This includes the value of certain traditional foods (e.g., pig, taro/dalo) which signify respect and hierarchy. For example:


*Certain foods are shared in social occasions or functions and have higher value such as ufi (dalo) and so these are shared or gifted with families and members as a sign of respect… For instance, different parts of fish or meat are given to certain important members of the family (usually the head or the eldest (O_Fiji).*


Food creates a ‘safe cultural space’ that provides opportunities for individuals to express themselves, reaffirming identity and a sense of belonging. Food unites cultural groups and communities. The sub-themes identified were food as: connection; a language of love; a way of honouring family and showing respect; a way to create culturally safe places; and as celebration. See Appendix A for exemplar quotes for sub-themes. The key role of traditional foods in keeping culture alive was highlighted: 


*We had one (underground oven) at the premises and one again at campsite. We have a number of youths [who] turned up and you can tell they thoroughly enjoyed the experience. This is how we cook back at home, how our ancestors cooked. It wasn’t with an electric oven that we turn the thing on. It was with wood and rocks. (O_Niue).*


### 3.2. Hospitality and Reciprocity

Food plays a central role in demonstrating the key cultural tenets of hospitality, reciprocity, respect and humility as illustrated by the themes described below. Food is part of a communal system connecting the Māori and Pasifika diaspora: an offer to share food may simply be an expression of hospitality and reciprocity guided by spoken and unspoken rules and demonstrates cultural appreciation. As with identity, there is a collective understanding that the main purpose of contributing food for shared activities is to enable participation and inclusion of all members of a community. In this context, contributing food is taking responsibility and maintaining cultural traditions within a cultural system of support to preserve social order and hierarchy. Often this means not only bringing food but also eating the food that is prepared. 


*Hospitality is a big thing in our church, because it sort of creates that sense of connectedness again. So even at the moment through COVID, it’s been a little bit tough, because we haven’t been able to do like big connection groups like or big corporate gatherings. But when we do when we are able to have them, it’s always kind of centred around food when we have that time to connect off afterwards (Y_ Māori).*



*Hospitality, you know, people who would visit us, the first thing we would provide is a drink. And then after that, if they were staying longer then we would provide food and have a meal, for example. But food seems to be the thing that brings everyone together and then people share that hospitality, share that fellowship, [and] share the social side of things (O_ Rotuma).*


Reciprocity was conceptualized as ‘nobody goes hungry’ and ‘there is always more than enough food’. Food is never offered sparingly as that would be deemed disrespectful and only the best is offered to guests and visitors. Whilst this can be detrimental to some families who are struggling, there is a communal culture of service with community members and families helping and supporting those in times of need.


*Well, I think that, you know, what, some families, we’re that that are not so well off, I think that is excellent, as well. terms of, I suppose that they have to make sacrifices and struggle a bit in some areas, because, you know, their country, they have to contribute to the community. But it’s like a lot other Pacific Island communities here is very kind of communal type, you know, communal system, I suppose it’s kind of like a bit of, you know, reciprocal behaviour that goes on as well. Because if you have a function as well, you know, for example, that the people from your church will help you out too… (O_ Niue).*


A formal practice of reciprocity is the tradition of ‘Inati’ (pronounced in-at-see), which is a shared concept practised for generations among Tokelauans, with similarities across other cultural groups. Inati implies that there will always be leftovers and plenty to go around. It means enough food has been provided and it is shared amongst everyone:


*Inati basically means the sharing of food. It’s contributing to each family …When they call the Inati, it is always boys. It’s not voiced for the mothers or the fathers. It has always been said that, for us with the “moya tomaniti” (meaning our children), we say there is Inati happening at two o’clock this afternoon. Okay, so each family will go, one or two people will go to that Inati. ….So basically, what they do is they calculate how many people in that family and then they’ll share accordingly. A family of three cannot have the same amount as a family of 10. So, they actually distribute accordingly (O_ Tokelau).*


While one side of reciprocity is in the giving, the other side is in accepting. To not accept what is offered is also considered disrespectful. However, some members of the younger generation believe this is changing.


*So in my role, I’ve done home visits, things like that, if somebody offers me food I won’t turn it down. It’s rude. Where I’m from, it’s rude. And I’m also seeing people who don’t have lots of money who are in crisis. And if they’re offering me something, I’ll take it, because they’re offering. That offer in itself means a lot when they haven’t got enough (M_ Māori).*



*I don’t think it’s not as, I guess, taboo to not eat someone’s food that they’ve prepared for you. I guess if they prepared it specifically for you, then yes, you should have a plate or at least eat from the plate. There’s no real expectation for you to finish it. But there is an expectation for you to keep serving (Y_ Kiribati).*


### 3.3. Spirituality

Spirituality plays a pivotal role in Māori and Pasifika communities and is observed through religious or faith practices as well as metaphysical, mental and physical connections to land and ancestors. Food is sacred and therefore connected to spirituality. Food that is grown, collected, prepared, shared and consumed acknowledges an ‘exchange of energy’ or ‘mana’ between all elements of the physical, metaphysical and social spaces and these connections are essential to holistic health. In Australia, families who have migrated have adapted to their new environments, moving away from growing their own foods. As a result, this has contributed to cultural and spiritual disconnect from the process of growing and appreciating this ‘exchange of energy through food and thus ‘feeding our mana’.


*We are very connected to the process of food, the way we work and toil the land, and so we appreciate food, you know. It’s all connected and for us Pasifika you know food and the land, we [have to] thank God. It’s a spiritual connection that we have to the land and how it gives back to us (M_Tonga).*



*I’d say mainly because it’s one way that we show our appreciation or love. Because we don’t live on monetary things, especially in the Islands. And a lot of it is because you put a lot of time and effort into growing it and cultivating it, and then into harvesting it. You’ve put all this time and energy into this and you’re consuming it. So, you’re consuming that energy for your body (Y_Kiribati).*


From a religious perspective, food is honouring God with the first fruits of one’s labour. Thus, sharing of food, for example, after a church service, affiliates with this purpose that ‘mana’ or strength comes from the ability to share and the ability to receive when one is vulnerable or hungry. 


*Not only the sacrament on Sunday, but the food that God provides for us every day is holy. And so therefore, we treat this as a provision that God has given to us to sustain our life, to develop us. Every holistic development of a human-being centres on this provision of God, and in your context this is food. So therefore, it is very important that food needs to be clean. Hygiene needs to come into all perspective of development, you know, gardening, growing, cooking, all those aspects; how to preserve them for the use of the family of the community, how to offer them in generosity to those who have come. And then they all are contained in God ‘Mana’.... that’s it (O_Rotuma).*


### 3.4. Expectations and Obligations

Food for Māori and Pasifika communities is also centred around fulfilling complex obligations to the community and church. Expectations could include contributing by bringing food to an event or gathering, by contributing financially or through participation that is, food preparation, cleaning and/or serving. Being present and actively contributing in some way is valued and is seen as important in maintaining relationships and connections. However, if these obligations or duties are not fulfilled, there may be negative consequences for families or individuals within their communities. While participation was considered a form of contribution often the types, quality and quantity of food provided were taken as the definitive benchmark of the value of the contribution. In many communities the contribution was judged against the social norms that are spoken and unspoken:


*I think there is definitely an expectation. I think there’s an expectation that if you want to be part of a community and being an active member, that everyone must contribute. So if everyone’s contributing food, then every member that turns up must contribute food. I think there’s that cultural expectation there. So I think for those who choose not to be involved, it could be a fact that they just financially aren’t able to contribute and then therefore cannot participate in those community events. And then that can kind of I guess impact our cultural identity, and not being able to be part of our cultural community can have such a massive impact (Y_Kiribati).*



*Mostly, because I give back. Yeah. So in my head, I’m like, it’s not something that I would expect. It’s something that I would do. More so out of obligation, rather than because I want to. Yeah. If that was the case, because it’s kind of that notion of you scratch my back and I’ll scratch yours. Which to me is like, I don’t love it but, again because I think when I give or when I do it, there is the expectation and intention of not receiving. It’s purely for the intention of giving (Y_Māori).*


There is intergenerational transfer of cultural knowledge through food which is fundamental to maintaining identity, connectedness and traditional practices. For diasporic communities, there are, however, changing priorities for younger generations in relation to financial commitments, family structure (nuclear instead of extended families), with mixed marriages of partners not necessarily from the same cultural background, obligations and work and family commitments. For many of the younger generation, there is a shift away from traditional practices and how food is perceived in terms of access, preparation, buying, feeding and consumption.


*I don’t feel like it should always be an expectation. I think if I knew that my brother or sister wasn’t working or were just making enough to provide for their own kids, that is expectation met. So it would be like if I’m hosting or having a big family dinner, the expectation is that they just turn up. I think it’s just about finding that balance, because I think some of the expectations are old school. Really, really old school. And for me, I guess, the way that I live now isn’t really applicable, or is not always applicable. So it’s hard to sort of follow them (Y_Māori).*



*But I think that it’s also dictated to by things such as changing priorities of young people, and the cost factor. For example, some young people, they just like a cup of tea, you know, some biscuits, sandwiches at their function… It is a lot easier to prepare. Usually when you’re providing a Pacific Island type meal, it takes a long time to prepare. It can be very costly as well. Some of the younger ones, you know, got mortgages, they’ve got kids or got sports fees, and so forth (O_Niue).*


#### 3.4.1. Expectations from Family 

For the Māori and Pasifika diaspora living in Australia and Aotearoa, it is common that expectations are placed on them to support extended families living in the same country or in their home Islands either financially or through other means. Pressures of not being able to fulfil such obligations can cause stress, conflict and discord within families and communities. It can also result in financial and material hardship for those living in Australia. 


*The view from the Islands with relatives is wealth but [they] do not realise that its more expensive to live here. I know there is an expectation for those of us overseas to contribute back home… One of the things that I’ve tried to do is explain to people back at home that some here and a significant number of Fijians and Pacific Islanders, they have to work two or three jobs to support their family. …. I know of people that are supporting their own families back home, they pay their electricity bill and everything else, including food.” (O_Fiji, Samoa, Cook Islands, Māori).*


#### 3.4.2. Expectations from Churches

Churches in many Māori and Pasifika communities are an integral part of the spiritual and social nourishment of members. Churches play a pivotal role in Pasifika communities, providing pastoral care that includes spiritual, mental, physical and financial support. Being well connected to communities, church can also play a key role in providing cultural support by providing safe spaces for Indigenous languages, traditional dress and practices to be expressed. Through the offering of charitable services, churches provide a significant safety net for those in need. It is also a space where trust is established; thereby creating a system of support for those in need. Families facing hardship, who are connected to the church, will use this resource before seeking external help. Knowing who needs help through the ‘coconut wireless’ (a community communication network) ensures that the support can be reciprocated in time. 


*I think the role of the churches in its pastoral care is important that we look at people who do not have and to send food or think how do we help them?… The money that God gives us as a gift can be channelled to other important things in the lives of the people because the church is a people. (O_Rotuma).*


However, different churches have different expectations of their members. For some participants, the expectations of church members to contribute equally, particularly to the financial viability of the organisation, potentially contributes to hardship and reduces the effectiveness of the safety net. This expectation was not necessarily experienced by the younger generation.


*I found out why a lot of people weren’t coming [to church]. Because they didn’t have the koha (donation) (O_Niue).*



*It can become a burden to the people that they are helping or an expectation from them to contribute… It’s a sad thing to say, but I think it’s more than the forcefulness of it… It’s like I need you in the church to show that our church is growing, that’s why they do what they do… (O_Samoa).*


With churches being the predominant safety net, there can be challenges for families or individuals who are not part of a church or faith-based community. This is often the case for many newly arriving Māori and Pasifika individuals or families who are unfamiliar with the networks and may not know who to go to for support or help. 


*Unfortunately, there’s also a lot of people that do not have that support. They don’t attend any churches, they pretty much like live on their own really and are not involved in any of their respective communities and cultural communities. So they’re the ones that will go without having family and social support around them (O_ Samoa).*


### 3.5. Stigma and Shame

There are two elements to this theme. The first is not being able to honour the expectations and obligations, outlined above, generating feelings of stigma, embarrassment and a sense of being judged. Families who are struggling may not be able to reciprocate or make financial or food contributions. Consequently, this could lead to not being able to connect to the wider community, resulting in further isolation and disengagement. Alternatively, the drive to honour these expectations can result in families going without food or using money set aside for rent or other core living expenses. 


*But I think in terms of anyone who shows up to an event where there’s no food, it will be looked down upon. And probably classified as rude. It could just be they just don’t have any money to be able to provide it and it’s kind of a no-win situation sometimes which I find it really harsh in our culture (M_Samoa).*



*It’s sad, I see this a lot. When families haven’t got anything to contribute, they stay home. Because sometimes it’s a shame to go to our traditional thing when we know everyone else will be putting in something, and I don’t have anything to put in. Because usually this time, they sometimes read out names. And if your name doesn’t get read out, it can be a very shameful experience. So the family doesn’t go. So what happens is more and more people don’t go because [of] their financial (issues) (O_Niue).*


The second element is a culture of pride within many Māori and Pasifika communities to hide hardship. Shame is associated with not having food. Keeping family matters private or using family safety nets is a way to maintain social identity and family honour. This strong culture can create barriers and challenges to seeking help or support or disclosing to others about issues of food insecurity. 


*In my experience, the majority of the time this struggle was never expressed outside the family. So in the same way, as the individual might not talk about, might not bring their struggle to the community space… Maybe it’s viewed as weakness, maybe as soon as they don’t want to impose on the space, or the community (Y_Samoa).*



*I think there’s still a lot … to learn, you know, when you’re doing it tough to put your hand up. You know, again, it comes down to that sense of pride, where you don’t want people to know or anyone else to know what you’re going through. (O_Cook Islands).*



*Because I think it’s a hard thing to go and ask for free food. Yeah, it’s shame. Men are supposed to be supporting your family providing for your family and you and you can’t even provide the basic of food on the table. I think it really took a kick in the gut kind of thing for them (Y_Tonga).*


For many Pasifika men not being able to put food on the table represents a loss of their manhood and a sense of failure that they are unable to provide for their families which can impact family dynamics and relationships as well as food security. This may not be an issue for those living in the Pacific Islands where food has been in abundance and was able to be collected, freely, from the land and sea. This was, however, identified as an issue for those living in Australia. 


*Well, I can speak on behalf of men. And I know that for any normal man that can’t put food on the table, it would be quite devastating. He would feel inadequate, and as if he wasn’t doing his job properly. Especially if there’s kids in the family. …. Because in Nauru, you don’t really have a problem with providing food on the table, because you can just go and catch fish. And yeah, you can get everything you need off the land and off the sea (O_Nauru).*


### 3.6. Physical and Mental Health

Health is an important issue for many Māori and Pasifika peoples living in Australia where there is a burden of chronic diseases and poor health status because of behaviours influenced by environmental (obesogenic environments—that is, easy access to energy-dense foods plus lack of time) and financial conditions. Participants identified feeding large families on small budgets, with adults working multiple jobs and with long hours, creating time poverty, as a particular issue. Providing cheap, energy-dense fast food was also considered an option for community events for those with fewer financial and time resources. Other key issues included lack of understanding about portion size, being mindful of eating healthily and the connections between food, nutrition and health. 


*So let’s say you have two parents, and a large family. Between them working, then looking after the kids then coming home? The question is like, Where’s the time to produce a good meal? Like where is the energy if they’re running between looking after, maybe there’s, a five-year-old and a sixteen-year-old, between them playing sports or juggling a lot of things? (Y_Samoa).*



*There may be the lack of knowledge and understanding the lack of skills to be able to know what to do with some of the different variety of foods. So one of the problems we have is too many portions, now we have too easy access. It’s not really appreciated, we just buy a whole bunch, and we’re eating copious amounts in every meal, because we went for the money and we’re just trading with money. And so because of that disconnect of that cycle of development, growth, distribution, evening out that cycle in the Islands is almost a little bit different now (M_Tonga).*


## 4. Discussion

Food security is an issue for the Māori and Pasifika diaspora living in south-east Queensland, but its ramifications go beyond the material provisioning of food and reach deeply into identity, belonging and community obligations. These in turn impact on health and wellbeing through disrupted relationships between food, environment, genealogical ties, cultural identity, resilience, reciprocity, respect and humanity [54,55]. The results from this research remind us that structural changes to improve access to and availability of healthy foods needs to take into consideration the expectations and obligations that bind communities together. The results highlight a range of points of discussion, but we want to focus on three key components; firstly, cultural food security is a salient consideration; secondly, that food security for Māori and Pasifika peoples is not an individual but a collective responsibility; and finally, food sovereignty approaches may provide more sustainable long-term solutions.

### 4.1. Cultural Food Security

Each domain of food security is influenced by culture from how food is grown, procured, prepared and shared [4]. The essentiality of cultural foods to food security acknowledges that cultural food security emphasizes the ability to reliably access important traditional foods, using traditional growing, harvesting and cooking techniques [43]. For many Māori and Pasifika peoples living in Australia, there is an accumulative impact of identifying as both Indigenous and migrant. Foodways and maintaining food security become imperative to building both a collective Pasifika identity as well as an individual Pacific Island nation identity. This may typically involve growing, preparing and sharing traditional foods in Australia, importing traditional foods from the Islands, maintaining to-and-fro movements between the diaspora and the Islands as well as the maintenance of these conduits by sending remittances and other support.

Food is one of the primary vehicles for social connectivity for the Māori and Pasifika diaspora. It is the materiality that underpins the spiritual, emotional, physical and social health of each member as well as the collective. Food is key to the building and maintenance of social capital [56]. Social capital (as social participation, networks and trust in others) is protective of migrant mental health, especially in those exposed to racism and discrimination [57,58] and can contribute to flourishing [59]. Flourishing is optimal functioning in emotional, psychological and social well-being—the opposite is languishing [59]. Cultural food security is therefore imperative for not only the maintenance of Maori and Pasifika community health, but is necessary to enable the communities to flourish.

Food insecurity in Māori and Pasifika communities is also contributing to the increased consumption of less healthy non-traditional options, foods that fill (are calorically dense) rather than nourish (are nutrient dense) [41,42]. There is a growing tension between the obligations and expectations associated with demonstrating hospitality and generosity with the need to limit portion sizes for health. The younger generations are providing leadership in finding compromises that fulfil community social obligations without compromising individual health. 

### 4.2. Food Insecurity as a Collective Responsibility 

The Food and Agriculture Organization (FAO) has recently extended the conceptualisation of food security to encompass two additional domains “agency” and “sustainability” [1]. The agency domain is relevant here in that it refers to the rights of individuals to determine their own food system. Agency as defined by Sen [60] is when a “person is free to do and achieve in pursuit of whatever goals and values her or she regards as important” (p. 203). While the domain stresses individuals and groups, the concept is still embedded in individualistic rather than collective conceptualisations. 

Individualistic cultures which are pervasive in most countries with neoliberal governments focus primarily on the care of self and family with less reliance on others for support [61,62]. Personal goals and identity take precedence over the goals of society and social identity. Conversely, collectivist cultures, which encompass many Indigenous peoples, focus on communal responsibility, interdependence and collective survival [63]. Harmony and relationships and the health of the whole community rather than individuals are prioritised. There are indications that food security (FS) at the household level may be viewed differently from the perspective of collectivist versus individualistic cultures. Renzaho and Mellor [64] describe food sharing and providing for needy families not as a coping strategy but rather as a “cultural obligation and a prerequisite for cultural harmony and community cohesion” (p. 2).

A strength of the Māori and Pasifika diaspora is the reliance on the collective in a complex interplay between expectations and obligations to ensure that all members of the community have access to food. For many communities, this is mediated by faith-based organisations that play a central role in ensuring well-being. Religious identity has been shown to contribute to psychological wellbeing directly and indirectly through social connectedness and social support [65]. These networks are responsible for ensuring that pride is maintained, so families in need do not need to ask for help. The outward manifestations of food insecurity in collectivist cultures range from not being able to meet your obligations and maintain social connectivity through food, to going hungry. Food insecurity is therefore not an individual failing but potentially a failure of the collective network. The stigma and shame that accompanies food insecurity is consequently potentially experienced by not just the individual but reflects on the entire community. Stigma and shame associated with food insecurity are not unique to Māori and Pasifika communities. They are experienced almost universally by those who are struggling to put food on the table, and in particular by those who have resorted to using charitable food relief [66,67]. Dryland et al. [68] describe the maintenance of social identity as a key driver for women experiencing food insecurity. In these situations, women prioritised strategies that provided a public demonstration that their situations were not compromised. For Māori and Pasifika families, this can mean maintaining their obligations in terms of church gift-giving, hospitality, reciprocity, remittances to family in the Islands and contributions at social functions. While at home they may be struggling to put food on the table.

The depth of distress experienced by Māori and Pasifika communities could potentially impact on how household food insecurity is measured. The FAO has recently started to move away from using only undernourishment as an indicator for food insecurity towards an experience scale (Food Insecurity Experience Scale (FIES)) [69]. The FIES is an eight-question tool based on the measure used in the United States and asks at the individual or household level about experiences of food insecurity (from worry to hunger) due to lack of money or other resources. The same measure is being used across countries to allow cross-country comparisons. There is no data for the Oceania region apart from Australia and New Zealand [70]. Given the impact of climate change in the region and the ensuing transformation of food systems, understanding the level of food insecurity experienced at the community level is vital [71]. The FAO in a personal communication indicated “we face an issue with fielding the FIES in some Pacific Island states, from where we receive reports that the questions are perceived as offensive by members of local communities and we are looking for means to better adapt them to culture and context to preserve their efficacy in eliciting the signs of food insecurity” (FAO, personal communication 23 January 2019). Developing the appropriate measure for monitoring will require understanding the broader ramifications of food insecurity and working with communities to identify and explore the issue in depth.

### 4.3. Moving to a Food Sovereingty Approach

The more recent conceptualisation of food security to include agency and sustainability as key dimensions highlights the possibility of moving towards a food sovereignty approach. Food sovereignty implies a broader vision of food security, moving away from an individualised focus to one that disrupts and reimagines food systems so they are democratically managed and geographically specific [72,73]. Food sovereignty has been conceptualised within First Nations contexts, as having seven pillars: (1) focuses on food for people; (2) builds knowledge and skills; (3) works with nature; (4) values food providers; (5) localizes food systems; (6) puts control locally; and (7) allows that food is sacred [74]. This approach aligns with the conceptualization of food for Māori and Pasifika communities living in south-east Queensland. That is, food is deeply linked to identity, spirituality, culture and holistic health. Food sovereignty highlights that the human right to food encompasses access to food-producing resources such as land and water, which can be more challenging in urban environments and for those disconnected from traditional resources via migration [75]. Food sovereignty requires an understanding and disruption of the inherent power structures and a focus on self-determination and social justice to enable individuals and communities to define their own food system [76]. Food sovereignty is described as a process as much about creating connectivity as about creating autonomy [77]. A food sovereignty approach also links more tightly with the Sustainable Development Goals with a focus on ensuring the health of land and seas, the maintenance of livelihoods and the empowerment and education of people [78].

Currently, instead of collective food sovereignty approaches, individualised interventions such as emergency food provisioning or cooking and budgeting workshops are more pervasive. These provide a valuable food and skills safety net but do not address long-term structural barriers. Health programs, to date, have not focused on community responsibilities but have instead “promoted individualised empowerment discourses which dissuade a symbiotic understanding of self-in-relation to community and land, place-based agency and social–ecological resilience” [79] (p. 57). Māori and Pasifika communities already have powerful social networks, deep knowledge and skills about food systems and a spiritual imperative to build a collective food network. What will be key in developing any strategies for Māori and Pasifika communities is recognition that the community is not a homogenous community but rather a multiplicity of cultural backgrounds and identities. There are also generational changes –with younger generations potentially less strongly connected to existing community safety nets that older generations have relied on and drawn support from. Māori and Pasifika youth in the diaspora are struggling and reconnection to cultural identity, spirituality and community through food may offer one pathway to self-determination [33,80]. 

This work has identified that future research needs to focus on co-designing solutions using a food sovereignty approach that will enable communities to flourish. The impact of social issues such as structural violence and racism are under-explored determinants of food insecurity as are potential protective factors such as community cohesion and religiosity. Finally, for Māori and Pasifika peoples, ensuring that food insecurity can be measured and monitored in a respectful way is a priority to enable adequate resources to realign food systems for sustainable action.

### 4.4. Limitations 

There are cultural differences between Māori and Pasifika communities and where possible these are highlighted but it has not been feasible in the context of this paper to explore the nuanced differences. The SC attempted to ensure representation from all Pacific Island nations, but we were unable to identify key informants from some of the smaller communities that may not be connected to the broader Pasifika diaspora. For example, there was no representation from the Marshall Islands or American Samoa. Some of the interviews took place at the height of the COVID-19 pandemic when many members of the community were in isolation. This situation had the potential of bringing food insecurity to the foreground due to the loss of jobs and income experienced by those who were ineligible for government payments. However, it also highlighted the resilience and strength of social networks that were mobilized to support community members. The data presented here is relevant within the context of Māori and Pasifika peoples living in south-east Queensland and may not be able to be generalized to the diaspora living in other locations. The conceptualization of food security as going beyond simple food provisioning for collectivist cultures could be relevant for other cultural groups.

## 5. Conclusions

Food has an important role in Māori and Pasifika communities in south-east Queensland and contributes significantly to collective identity, spirituality and holistic health. There are complex obligations and expectations involving food that are necessary for the maintenance of relationships and community. When foods (including cultural foods) are not available or accessible, individuals experience not only a loss of physical health but lose connections to their broader identity and their link to their ancestral homes. However, communities have demonstrated resilience through the maintenance of social, spiritual and cultural support networks. Churches and local community groups provide culturally safe and far-reaching safety nets for Māori and Pasifika families and individuals struggling to put food on the table. 

Addressing food insecurity for collectivist cultures such as those of Māori and Pasifika peoples requires reframing its definition and scope to embrace food sovereignty approaches over individualised or nuclear family-focused methods. It requires a holistic appreciation of cultural food security whereby elements of identity, reciprocity, hospitality, mana and spirituality are all regarded as significant and interconnected. In this context, improved food security may be attained through co-creation and co-designing solutions which range from high-level actions that impact the social determinants of Māori and Pasifika peoples as well as practical solutions that utilise and strengthen existing social networks to produce and distribute affordable and nutritious food. 

## Figures and Tables

**Figure 1 nutrients-14-02020-f001:**
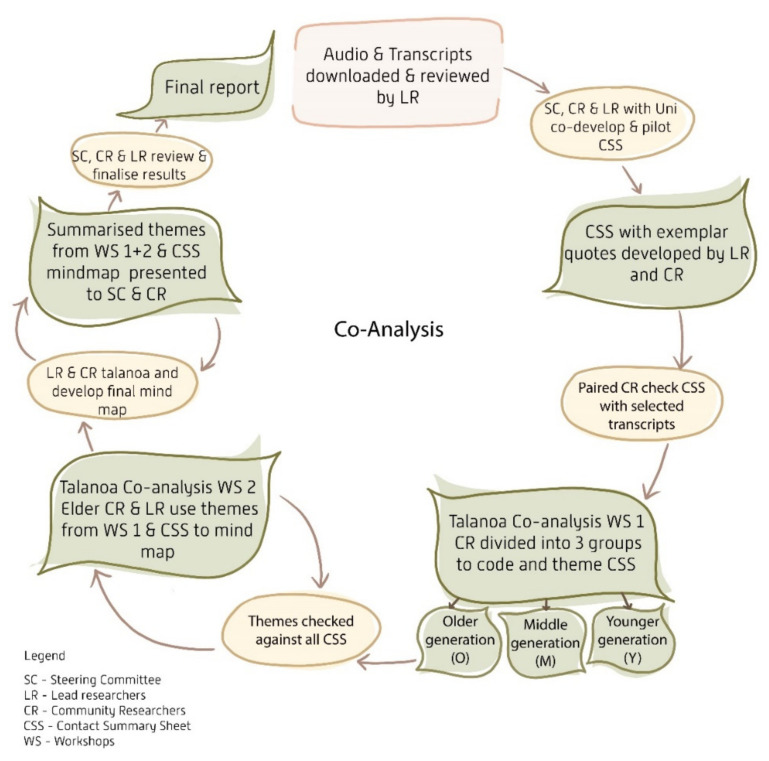
Data analysis process.

**Figure 2 nutrients-14-02020-f002:**
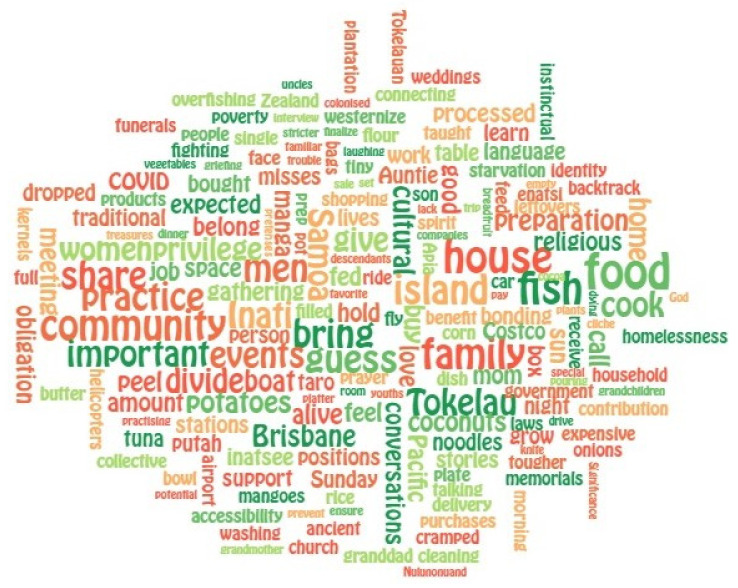
Wordle example from one of the interviews.

**Figure 3 nutrients-14-02020-f003:**
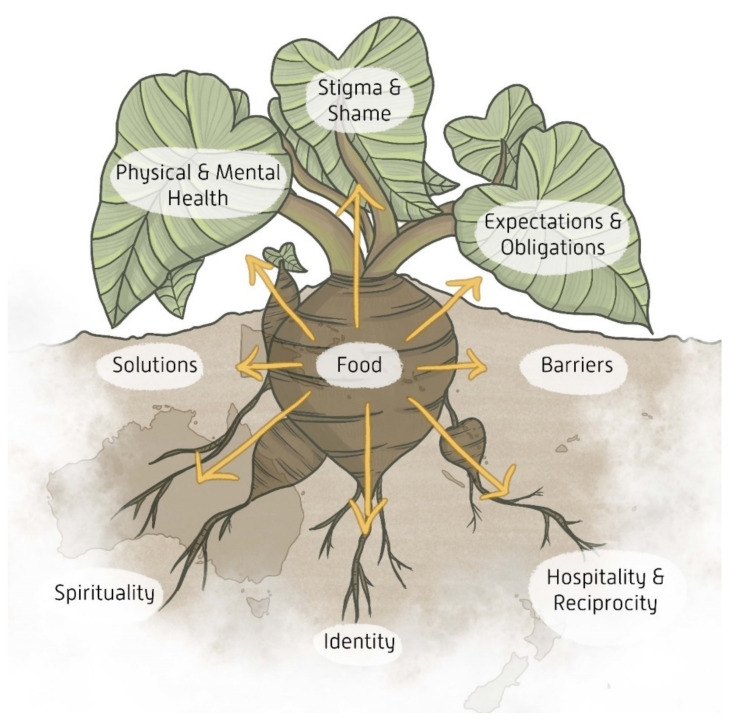
The taro plant illustrates eight key themes that emerged from the interviews which are shown to be all interconnected by food: identity, hospitality and reciprocity, and spirituality form the roots or foundation of the meaning of food for Māori and Pasifika peoples; solutions and barriers form the soil or structure from which these understandings and practices of food are nourished or restricted; physical and mental health, expectations and obligations, and stigma and shame form the leaves of the taro plant, representing the surface-level experiences and challenges of food insecurity. The diagram overlies a map of Australia, Aotearoa and some Pacific Islands to symbolise how our peoples’ experiences of food insecurity may differ between diasporic and homeland communities but are simultaneously intertwined by socio-cultural and genealogical ties that transcend space and time.

**Table 1 nutrients-14-02020-t001:** Cultural identity as reported by participants in the interviews *.

Cultural Identity	Participants
Cook Islands	1
Federated States of Micronesia	1
Fiji	5
Kiribati	1
Kiribati/Australia	1
Māori/Pakeha	2
Māori	5
Māori/Cook Islands	1
Nauru/Australia	1
Niue	3
Papua New Guinea	4
Rotuma	1
Samoa	7
Samoa/Fiji	1
Samoa/Māori/Irish	1
Solomon Islands	2
Tokelau/Māori/Tuvalu/Cook Islander	2
Tokelau	2
Tonga	4
Tonga/Australia	2
Tuvalu	1
French Polynesian-Tahiti/Māori	1
Total	49

* We have reported the cultural identities as described by the participants. These include self-identification with a singular culture and with hyphenated identities dependent on the diasporic journey.

## Data Availability

De-identified data is available on reasonable request to the researchers.

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
