# Peer review of "“Food Is Our Love Language”: Using Talanoa to Conceptualize Food Security for the Māori and Pasifika Diaspora in South-East Queensland, Australia"

_nutrients, 2022, doi:10.3390/nu14102020_

Round 1

Reviewer 1 Report

I reviewed the manuscript entitled, ”Food is Our Love Language”: Conceptualizing Food Security for Māori and Pasifika Diaspora in South-East Queensland, Australia. What describes the Food is Our Love Language? Authors must revise the title to reflect the content related to food security in South-East Queensland, Australia. Although the manuscript is well written, there are few experimental errors with small sample size.

Line 18: co-design26ed.. What is this?

Line 20: sentences should not start with a number. 40 interviews… please revise it

Line 21: South-east, Is S capital letter or small letter? Please fix to one throughout the manuscript

Abstract should be revised with clear aims, research findings, and concluding remarks

Line 34: (p. 7)…. Can be removed

Introduction is too long. Authors should consider to concise highlighting the need of conducting this study

Lines 173 and 174: took place is not a technical word. I suggest that data collection was performed ……

Section 2.2: how many participants? this should be mentioned clearly. 

Results:

Lines 261-262: Very small sample size for this kind of study. How can authors define accuracy of research with small sample size?

Section 4.4: sample size is not a limitation?

Conclusions should be revised according to the research findings.

None of the references are according to journal format. Almost all references are inconsistent. Authors must follow journal guidelines and revise accordingly.

Reviewer 2 Report

Comment for authors

The present study entitled “Food is Our Love Language”: Conceptualizing Food Security 2 for Māori and Pasifika Diaspora in South-East Queensland, 3 Australia” is interesting but limited to a specific community and limited to the readership of nutrients. In addition, it only reflects the thought and perception of people. There are some other points that should also be addressed.

  1. Line 17: typos “design26ed”
  2. Aims of study are not clear. Please elaborate more in details. Scope of study and significance of study.
  3. Lines 260-268: this part seems to be a methodology.
  4. Authors should represent and explain more results of interviews. There is good explanation but the interpretation of interview is lacking throughout the manuscript.
  5. Write conclusion with respect to the finding of the study. I would suggest first present findings of study and then add concluding remarks based on results.

Round 2

Reviewer 1 Report

The authors are now included the suggestions made by me. In my opinion, the manuscript can be accepted for publication. 

Author Response

Please see attachment for updated manuscript
